

# Structure and stability of recombinant bovine odorant-binding protein: II. Unfolding of the monomeric forms

Olga V. Stepanenko[1], Denis O. Roginskii[1], Olesya V. Stepanenko[1], Irina M. Kuznetsova[1], Vladimir N. Uversky[1,2] and Konstantin K. Turoverov[1,3]

[1] Laboratory of Structural Dynamics, Stability and Folding of Proteins, Institute of Cytology, Russian Academy of Sciences, St. Petersburg, Russia
[2] Department of Molecular Medicine and USF Health Byrd Alzheimer's Research Institute, Morsani College of Medicine, University of South Florida, Tampa, FL, United States
[3] Peter the Great St. Petersburg Polytechnic University, St. Petersburg, Russia

## ABSTRACT

In a family of monomeric odorant-binding proteins (OBPs), bovine OBP (bOBP), that lacks conserved disulfide bond found in other OBPs, occupies unique niche because of its ability to form domain-swapped dimers. In this study, we analyzed conformational stabilities of the recombinant bOBP and its monomeric variants, the bOBP-Gly121+ mutant containing an additional glycine residue after the residue 121 of the bOBP, and the GCC-bOBP mutant obtained from the bOBP-Gly121+ form by introduction of the Trp64Cys/His155Cys double mutation to restore the canonical disulfide bond. We also analyzed the effect of the natural ligand binding on the conformational stabilities of these bOBP variants. Our data are consistent with the conclusion that the unfolding-refolding pathways of the recombinant bOBP and its mutant monomeric forms bOBP-Gly121+ and GCC-bOBP are similar and do not depend on the oligomeric status of the protein. This clearly shows that the information on the unfolding-refolding mechanism is encoded in the structure of the bOBP monomers. However, the process of the bOBP unfolding is significantly complicated by the formation of the domain-swapped dimer, and the rates of the unfolding-refolding reactions essentially depend on the conditions in which the protein is located.

Corresponding authors
Vladimir N. Uversky,
vuversky@health.usf.edu
Konstantin K. Turoverov,
kkt@incras.ru

## INTRODUCTION

Odorant binding proteins (OBPs) are important components of olfactory apparatus in vertebrates where they play a specific role in olfaction by interacting directly with odorants (*Xu et al., 2005*). OBPs constitute a class of small extracellular proteins in the chemosensory systems of most terrestrial species ranging from drosophila to human. In mammals, OBPs are found at high concentrations (∼10 mM) in nasal mucosa of

cow (*Bignetti et al., 1985*; *Pelosi, Baldaccini & Pisanelli, 1982*), rat (*Pevsner et al., 1985*), rabbit (*Dal Monte et al., 1991*), pig (*Dal Monte et al., 1991*), dog (*D'Auria et al., 2006*), and humans (*Briand et al., 2002*). Although they bind different kinds of small and hydrophobic odorant molecules (typically with the affinities in the micromolar range, their inability to discriminate different chemical classes of these molecules suggests that OBPs cannot serve as olfactory receptors (*Boudjelal, Sivaprasadarao & Findlay, 1996*). The precise biological functions of mammalian OBPs are not known as of yet, but it was hypothesized that these proteins can be involved in transport of hydrophobic odorants across the aqueous mucus layer to access the olfactory receptors, or might be involved in the termination of the olfactory signal by removing odorants from the receptors after their stimulation (*Bignetti et al., 1987*; *Pevsner & Snyder, 1990*).

OBPs constitute a sub-class of lipocalins, which are small extracellular proteins found in gram negative bacteria, plants, invertebrates, and vertebrates. Although lipocalins are known to share limited regions of sequence homology, they do have a common tertiary structure architecture (*Flower, North & Sansom, 2000*; *Grzyb, Latowski & Strzalka, 2006*). The characteristic structural signature of the lipocalin family is a $\beta$-barrel composed by a 9-stranded anti-parallel $\beta$-sheet with an $\alpha$-helical segment at the C-terminus (*Flower, North & Sansom, 2000*). The internal cavity of the lipocalin $\beta$-barrel is the binding site that can interact with the odorant molecules belonging to different chemical classes (*Vincent et al., 2000*). Bovine OBP (bOBP) was the first OBP for which crystal structure was solved, and the analysis of this structure revealed that bOBP exists as a domain-swapped dimer (*Bianchet et al., 1996*; *Tegoni et al., 1996*). This was in contrast to structures of other lipocalins, including the porcine OBP (pOBP) (*Spinelli et al., 1998*), which are monomeric proteins. The ability of bOBP to form domain-swapped dimers was explained by the absence of a glycine residue at the hinge region linking the $\beta$-barrel to the $\alpha$-helix, and by the lack of the disulfide bridge which is present in all lipocalin sequences identified so far (*Ramoni et al., 2002*).

This work is dedicated to the analysis of the peculiarities of the GdnHCl-induced unfolding—refolding reactions of the recombinant bOBP and its monomeric mutants, bOBP-Gly121+ and bOBP-Gly121+/W64C/H155C (GCC-bOBP). It continues a series of articles dedicated to the analysis of the effect of the environment (including the presence of crowding agents) on structural properties and conformational stability of bOBP. The mutant protein bOBP-Gly121+ contains an extra glycine residue introduced after the bOBP residue 121. This substitution was shown to promote monomerization of the bOBP (*Stepanenko et al., 2016*) likely via increasing the mobility of the loop connecting $\alpha$-helix and 8th $\beta$-strand of the $\beta$-barrel. Substitutions of the residues Trp64 and His156 to cysteines in bOBP-Gly121+ generate a mutant form GCC-bOBP, which is expected to have stable monomeric structure due to the restoration of the disulfide bond typically seeing in other OBPs (*Ramoni et al., 2008*). We also investigated the role of the natural ligand in the stabilization of protein structure and looked at how the ligand binding affected the folding-unfolding reaction of these proteins.

## MATERIAL AND METHODS

### Materials

GdnHCl (Nacalai Tesque, Japan), 1-octen-3-ol (OCT; Sigma-Aldrich, USA) and ANS (ammonium salt of 8-anilinonaphtalene-1-sulfonic acid; Fluka, Switzerland) were used without further purification. The protein concentration was 0.1–0.2 mg/ml. The OCT concentration was 10 mM. The experiments were performed in 20 mM Na-phosphate-buffered solution at pH 7.8.

### Gene expression and protein purification

The plasmids pT7-7-bOBP which encodes bOBP and its mutant forms with a poly-histidine tag were used to transform *Escherichia coli* BL21(DE3) host (Invitrogen) (*Stepanenko et al., 2014c*). The protein expression was induced by incubating the cells with 0.3 mM of isopropyl-beta-D-1-thiogalactopyranoside (IPTG; Fluka, Switzerland) for 24 h at 37 °C. The recombinant protein was purified with Ni+-agarose packed in HisGraviTrap columns (GE Healthcare, Sweden). The protein purity was determined through SDS-PAGE in 15% polyacrylamide gel (*Laemmli, 1970*).

### Fluorescence spectroscopy

Fluorescence experiments were performed using a Cary Eclipse spectrofluorometer (Varian, Australia) with microcells FLR ($10 \times 10$ mm; Varian, Australia). Fluorescence lifetime were measured using a "home built" spectrofluorometer with a nanosecond impulse (*Stepanenko et al., 2014a*; *Stepanenko et al., 2012*; *Turoverov et al., 1998*) as well as micro-cells (101.016-QS $5 \times 5$ mm; Hellma, Germany). Tryptophan fluorescence in the protein was excited at the long-wave absorption spectrum edge ($\lambda_{ex} = 297$ nm), wherein the tyrosine residue contribution to the bulk protein fluorescence is negligible (*Stepanenko et al., 2015*). The fluorescence spectra position and form were characterized using the parameter $A = I_{320}/I_{365}$, wherein $I_{320}$ and $I_{365}$ are the fluorescence intensities at the emission wavelengths 320 and 365 nm, respectively (*Turoverov & Kuznetsova, 2003*). The values for parameter $A$ and the fluorescence spectrum were corrected for instrument sensitivity. The tryptophan fluorescence anisotropy was calculated using the equation: $r = (I_V^V - G I_H^V)/(I_V^V + 2 G I_H^V)$, wherein $I_V^V$ and $I_H^V$ are the vertical and horizontal fluorescence intensity components upon excitement by vertically polarized light. $G$ is the relationship between the fluorescence intensity vertical and horizontal components upon excitement by horizontally polarized light ($G = I_V^H/I_H^H$), $\lambda_{em} = 365$ nm (*Turoverov et al., 1998*). The fluorescence intensity for the fluorescent dye ANS was recorded at $\lambda_{em} = 480$ nm ($\lambda_{ex} = 365$ nm).

### GdnHCl-induced unfolding

Protein unfolding was initiated by manually mixing the protein solution (40 µL) with a buffer solution (510 µL) so that the GdnHCl concentration was varied from 0.0 to 4.0–5.0 M in the absence or presence of a natural ligand, 1-Octen-3-ol (OCT). The GdnHCl concentration was determined by the refraction coefficient using an Abbe refractometer (LOMO, Russia; (*Pace, 1986*)). The dependences of different fluorescent characteristics

of the studied proteins on GdnHCl were recorded following protein incubation in a solution with the appropriate denaturant concentration at 4 °C for different times (see in the text). The protein refolding was initiated by diluting the pre-denatured protein (in 3.0 M GdnHCl, 40 μL) with the buffer or denaturant solutions at various concentrations (510 μL). The spectrofluorimeter was equipped with a thermostat that holds the temperature constant at 23 °C.

## Fitting of denaturation curves

The equilibrium dependences of the fluorescence intensity at 320 nm on the GdnHCl concentration were fit using a two-state model:

$$S = \frac{S_N + S_U K_{N-U}}{1 + K_{N-U}}, \tag{1}$$

$$K_{N-U} = \exp\left(\frac{-\Delta G^0_{N-U} + m_{N-U}[D]}{RT}\right), \tag{2}$$

$$K_{N-U} = F_U / F_N = (1 - F_N)/F_N, \tag{3}$$

taking into account

$$S_N = a_N + b_N[D], \tag{4}$$

$$S_U = a_U + b_U[D], \tag{5}$$

where $S$ is the fluorescence intensity at the measured GdnHCl concentration; $[D]$ is the guanidine concentration; $m$ is the linear dependence of $\Delta G_{N-U}$ on the denaturant concentration; $\Delta G^0_{N-U}$ is the free energy of unfolding at 0 M denaturant; $F_N$ and $F_U$ are the fractions of native and unfolded molecules, respectively; and $S_N$ and $S_U$ are the signal of the native and unfolded states, respectively; $a_N$, $b_N$, $a_U$ and $b_U$ are constants needed to fit linear dependences of the $S_N$ and $S_U$ signals on the GdnHCl concentration. Fitting was performed using a nonlinear regression with Sigma Plot.

To evaluate conformational stability of the studied proteins we took into account that the formation of the native dimeric state of bOBP occurred at moderate GdnHCl concentration is followed by full protein unfolding while conformational perturbations of bOBP at low denaturant concentrations were not attributed to the unfolding of the protein globule (see "Results and Discussion" section). As bOBP unfolding is fully reversible the transition from native to unfolded state of the protein was used to calculate $\Delta G_{N-U}$ value. Conformational stability of the bOBP mutant forms was evaluated similarly taking into account that both mutant proteins unfold through the same scheme as bOBP does (see "Results and Discussion" section).

## Circular dichroism measurements

The CD spectra were generated using a Jasco-810 spectropolarimeter (Jasco, Japan). Far-UV CD spectra were recorded in a 1-mm path length cell from 260 nm to 190 nm with a 0.1 nm step size. For the spectra, we generated 3 scans on average. The CD spectra for the appropriate buffer solution were recorded and subtracted from the protein spectra.

## Gel filtration experiments

We performed gel filtration experiments for recombinant bOBPwt and its mutant forms in a buffered solution and with addition of GdnHCl using a Superdex-75 PC 3.2/30 column (GE Healthcare, Sweden) and an AKTApurifier system (GE Healthcare, Sweden). The column was equilibrated with the buffered solution or GdnHCl at the desired concentration, and 10 μl of the protein solution prepared under the same conditions was loaded on the pre-equilibrated column. The change in hydrodynamic dimensions for the studied proteins was evaluated as a change in the protein elution volume. Multiple proteins with known molecular masses (aprotinin (6.5 kDa), ribonuclease (13.7 kDa), carbonic anhydrase (29 kDa), ovalbumin (43 kDa) and conalbumin (75 kDa), which are chromatography standards from GE Healthcare) were used to generate the calibration curve.

# RESULTS AND DISCUSSION

## Structural features of the monomeric and dimeric bOBPs

Figure 1 compares structural features of the natural bOBP and its monomeric mutant GCC-bOBP. Structures shown in Figs. 1A and 1B indicate that the domain swapping has been reverted in monomeric GCC-bOBP. Furthermore, Fig. 1C represents the results of the multiple structural alignment of the one of the monomers of the natural dimeric bOBP (blue structure), monomeric mutant GCC-bOBP (red structure), and naturally monomeric pOBP (green structure) and clearly shows that the monomeric GCC-bOBP has the overall fold almost identical to that of the pOBP. Next, we analyzed how introduced mutations affected the structural flexibility of bOBP utilizing the power of the FlexPred tool that rapidly predicts absolute per-residue fluctuations from a three-dimensional structure of a query protein (*Jamroz, Kolinski & Kihara, 2012*). Results of this analysis are shown in Fig. 1D which clearly indicates that the chains of the bOBP dimer, chains of the naturally monomeric pOBP and the monomeric mutant GCC-bOBP are all characterized by very similar structural flexibility. Since the FlexPred tool provides real-value fluctuations of globular proteins based on their static structures by considering Cα atoms contact number and known B-factor values, the fact that these three proteins, bOBP (PDB ID: 1OBP), GCC-bOBP (PDB ID: 2HLV), and pOBP (PDB ID: 1A3Y), are predicted to have similar structural flexibility indicates that mutations introduced to generate the monomeric GCC-bOBP or present in the naturally monomeric pOBP do not dramatically affect Cα atoms contact number and B-factors of the resulting structures. Note that these observations are in a good agreement with the overall very high structural similarity of these three proteins, the observation supported by the fact that the multiple structural alignment of these three proteins over 136 residues was characterized by the RMSD of 0.89 Å (see Fig. 1C). We also compared these results of structure-based flexibility of the bOBP with the propensity of this protein for intrinsic disorder evaluated by PONDR® VSL2B, which is one of the more accurate stand-alone disorder predictors (*Fan & Kurgan, 2014*; *Peng et al., 2005*; *Peng & Kurgan, 2012*). Results of this analysis are also shown in Fig. 1D which illustrates that there is generally a very good agreement

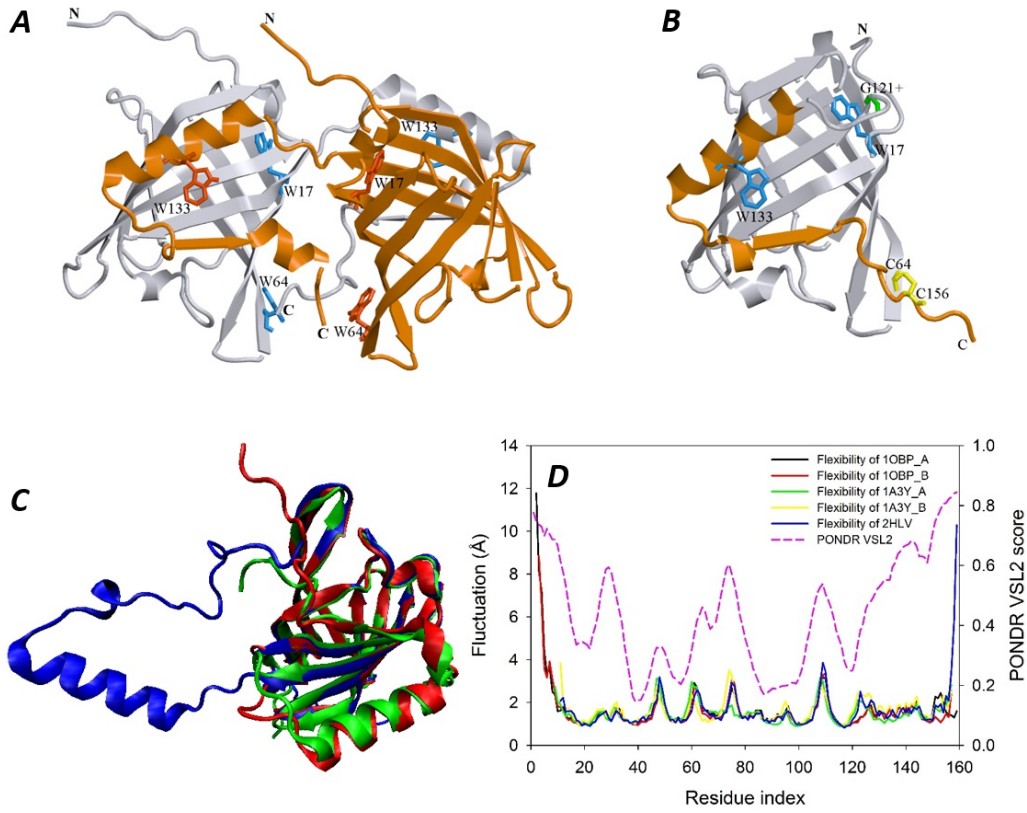

**Figure 1  Analysis of the 3D structure of bOBP.** Crystal 3D structures of natural bOBP (A) and monomeric mutant form GCC-bOBP (B). The individual subunits in the bOBP are in gray and orange. In the GCC-bOBP short $\alpha$-helical segment that followed by the 9th $\beta$-strand and the disordered C-terminal region of the protein are drawn in orange. The tryptophan residues are indicated in red and blue in the different subunits of bOBP and in blue in GCC-bOBP. The Gly 121+ residue which donates the increased mobility of the loop connecting $\alpha$-helix and 8th $\beta$-strand of the $\beta$-barrel and promotes the formation of a monomeric fold of the mutant protein bOBP-Gly121+ is in green. Two cysteines residues Cys 64 and Cys 156 in GCC-bOBP, which are believed to stabilize monomeric structure due to the disulfide bond formation are in yellow. The drawing of bOBP and GCC-bOBP was generated based on the 1OBP (*Tegoni et al., 1996*) and 2HLV files (*Ramoni et al., 2008*) from PDB (*Dutta et al., 2009*) using the graphic software VMD (*Hsin et al., 2008*) and Raster3D (*Merritt & Bacon, 1977*). Plot (C) represents the results of the multiple structural alignment of bOBP (PDB ID: 1OBP, blue structure), GCC-bOBP (PDB ID: 2HLV, red structure), and naturally monomeric pOBP (PDB ID: 1A3Y, green structure) using the MultiProt algorithm (http://bioinfo3d.cs.tau.ac.il/MultiProt/) (*Shatsky, Nussinov & Wolfson, 2004*). The drawing was generated using the graphic software VMD (*Hsin et al., 2008*). Plot (D) compares flexibility profiles obtained from crystal structures of bOBP (PDB ID: 1OBP, black and red lines for the chains A and B), naturally monomeric pOBP (PDB ID: 1A3Y, green and yellow lines for the chains A and B) and monomeric mutant GCC-bOBP (PDB ID: 2HLV, blue line) with the intrinsic disorder propensity of the bOBP (UniProt ID: P07435, pink dashed line). Flexibility profiles were obtained using the FlexPred software available at http://kiharalab.org/flexPred/ (*Jamroz, Kolinski & Kihara, 2012*), whereas intrinsic disorder propensity was evaluated using the PONDR® VSL2 algorithm (*Peng et al., 2005*).

between the structural flexibility calculated from the protein crystal structure and the propensity of a protein for intrinsic disorder.

## Equilibrium unfolding of the recombinant bOBP in the presence of natural ligand

Previously we have shown that the recombinant bOBP, unlike the natural bOBP extracted from the tissues, represents a mixture of monomeric and dimeric forms suggesting that this protein exists in a stable native-like state with a reduced dimerization capability (*Stepanenko et al., 2014c*). It has been suggested that the recombinant form of bOBP is characterized by the disturbed package of the $\alpha$-helical region and some $\beta$-strands, which prevents the formation of a native domain-swapped dimer (*Stepanenko et al., 2014c*). It is likely that the dimer formation via the domain exchange mechanism is a rather complex process that requires specific organization of the secondary and tertiary structure within the monomers. Curiously, recombinant bOBP can form the compact dimeric state under the mild denaturing conditions, namely, in the presence of 1.5 M guanidine hydrochloride (GdnHCl) (*Stepanenko et al., 2014c*). This process requires bOBP restructuring and is accompanied by the formation of a stable, more compact, intermediate state that is maximally populated at 0.5 M GdnHCl. Noticeably, at GdnHCl concentrations lower than 1.6 M we observed moderate changes in bOBP intrinsic fluorescence and far-UV CD spectra. These changes are not attributed to the bOBP unfolding process but are determined by some local structural changes in the protein globule. The dimeric bOBP in the presence of 1.5 M GdnHCl is characterized by a highly ordered secondary structure and a highly rigid microenvironment around the tryptophan residues. In the absence of GdnHCl, the recombinant bOBP is in a stable state with features similar to the native dimeric bOBP. Still, recombinant bOBP in the absence of GdnHCl is characterized by a less ordered secondary structure compared to the wild-type bOBP crystallographic data and a more rigid microenvironment of tryptophan residues, which is characterized by a decreased capacity of the recombinant bOBP for dimerization in aqueous solutions. This stable state of the recombinant bOBP state was designated as a ''trapped'' conformation with incorrect packing of $\alpha$-helices and $\beta$-sheet within the protein globule, which may interfere with the formation of the bOBP native state. The reasons for accumulation of this ''trapped'' state may lie in a relatively complex domain-swapping dimerization mechanism which is also required for the monomers to be correctly folded. On the other hand, the intermediate state of bOBP structure which is accumulated at 0.5 M GdnHCl is characterized by the reorganized bOBP structure that has fewer elements of ordered secondary structure, compared with the recombinant bOBP structure both in an aqueous solution and in the solution containing 1.5 M GdnHCl. The increase of the GdnHCl concentration above 1.5 M induces cooperative unfolding of the recombinant bOBP, which is completed by $\sim$3 M GdnHCl and is indicated by the simultaneous changes of all structural parameters of bOBP analyzed in this study (*Stepanenko et al., 2014c*). The half-transition point of this unfolding process at >2 M GdnHCl indicates high conformational stability of the recombinant bOBP (*Stepanenko et al., 2014c*), which is comparable with the stabilities of the native (isolated from tissue) bOBP (*Mazzini et al., 2002*) and pOBP

**Table 1** Hydrodynamic dimensions of recombinant bOBPwt and its mutant forms in the absence and in the presence of natural ligand OCT in different structural states.

|  | GdnHCl, M | First peak, kDa | Second peak, kDa |
| --- | --- | --- | --- |
| bOBPwt | 0.0 | 43.9 | 23.8 |
|  | 0.5 | 34.0 | 19.3 |
|  | 1.5 | 43.6 |  |
| bOBPwt/OCT | 0.0 | 39.6 | 21.5 |
|  | 0.55 | 27.2 | 17.0 |
|  | 1.7 | 39.6 |  |
| bOBP/Gly121+ | 0.0 | 23.6 |  |
|  | 0.25 | 17.8 |  |
|  | 1.5–1.9 | 24.8–28.5 |  |
| bOBP/Gly121+/OCT | 0.0 | 21.5 |  |
|  | 0.24–0.5 | 15.5–16.2 |  |
|  | 1.7–2.0 | 24.7–28.5 |  |
| GCC-bOBP | 0.0 | 23.6 |  |
|  | 0.25 | 17.8 |  |
|  | 1.1–1.82 | 22.5–25.9 |  |
| GCC-bOBP/OCT | 0.0 | 22.5 |  |
|  | 0.27 | 16.9 |  |
|  | 1.5–2.0 | 22.5–27.2 |  |

(*Staiano et al., 2007*; *Stepanenko et al., 2008*) and is inherent to $\beta$-rich proteins (*Stepanenko et al., 2012*; *Stepanenko et al., 2013*; *Stepanenko et al., 2014b*). We have also established that the unfolding of recombinant protein is a completely reversible process, whereas the preceding process of the dimerization of recombinant bOBP is the irreversible event (*Stepanenko et al., 2014c*).

To understand how interaction of the recombinant bOBP with its natural ligand, 1-Octen-3-ol (OCT) affects this protein, we investigated structural properties and conformational stability of the recombinant bOBP in the presence and absence of OCT. The formation of the protein-ligand complex does not affect the oligomeric status of the protein, since according to the gel filtration analysis the protein in the bOBP/OCT complex continue to exist as a mixture of monomeric and dimeric molecules. However, both monomeric and dimeric forms of the protein become more compact as a result of the OCT binding (Table 1, Fig. 2). The GdnHCl-induced unfolding of the bOBP/OCT complex is a rather slow process, since the equilibrium unfolding curves are achieved after incubation of the bOBP/OCT in the presence of different concentrations of the denaturing agent for more than 24 h. However, the established equilibrium was not affected by further incubation for up to 5 days (Fig. 3).

Figure 3 shows that the complexity of the bOBP/OCT unfolding is clearly illustrated by the complex shapes of the equilibrium dependencies of various characteristics of this complex on GdnHCl concentration. This suggests the accumulation of several intermediate states, which are similar to partially folded species found during the equilibrium

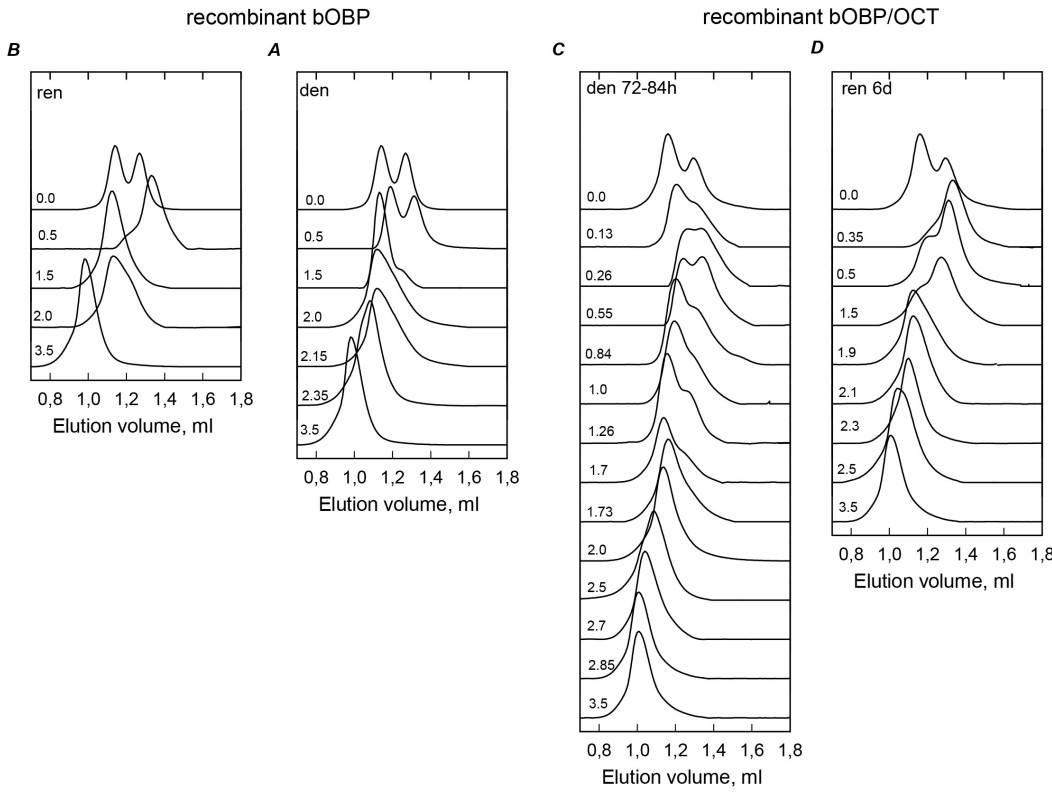

**Figure 2** **The changes of hydrodynamic dimensions of recombinant bOBP (A and B) and its complex with ligand bOBP/OCT (C and D) in different structural states.** The elution profiles for bOBP and bOBP/OCT were recorded during the protein denaturation (A and C) and renaturation from unfolded states (B and D) induced by GdnHCl. The elution profiles for bOBP were measured after pre-incubation of the protein and the solution of GdnHCl in desired concentration for 24 h (A and B), while in the case of bOBP/OCT the incubation time was extended to 72–84 h for denaturation (C) and 6 days for renaturation (D). The figures on the curves are the GdnHCl concentrations.

unfolding of the recombinant bOBP in the absence of OCT. However, compared to the bOBP alone, in the case of the unfolding of bOBP/OCT complex, the accumulation of these intermediate states takes place at higher denaturant concentrations. In fact, a more compact intermediate state of the bOBP/OCT complex is formed in the concentration range of 0.26–1.0 M GdnHCl, whereas the transition of the bOBP/OCT complex to the dimeric state occurs only at 2.0 M GdnHCl (Fig. 2). Increasing the GdnHCl concentration over 2.0 M leads to the cooperative unfolding of the bOBP/OCT complex. In comparison with the unfolding of the recombinant bOBP alone, the unfolding of the bOBP/OCT complex occurs in a narrow concentration range, and higher denaturant concentrations are required for complete unfolding of this complex. All these data indicate that formation of the bOBP/OCT complex leads to the substantial stabilization of the protein, but does not affect its unfolding mechanism.

To better understand the GdnHCl-induced unfolding of the bOBP/OCT complex, we analyzed the dependence of the 8-anilinonaphthalene-1-sulfonic acid (ANS, also called 1-anilino-8-naphthalenesulfonate), fluorescence on the denaturant concentration.

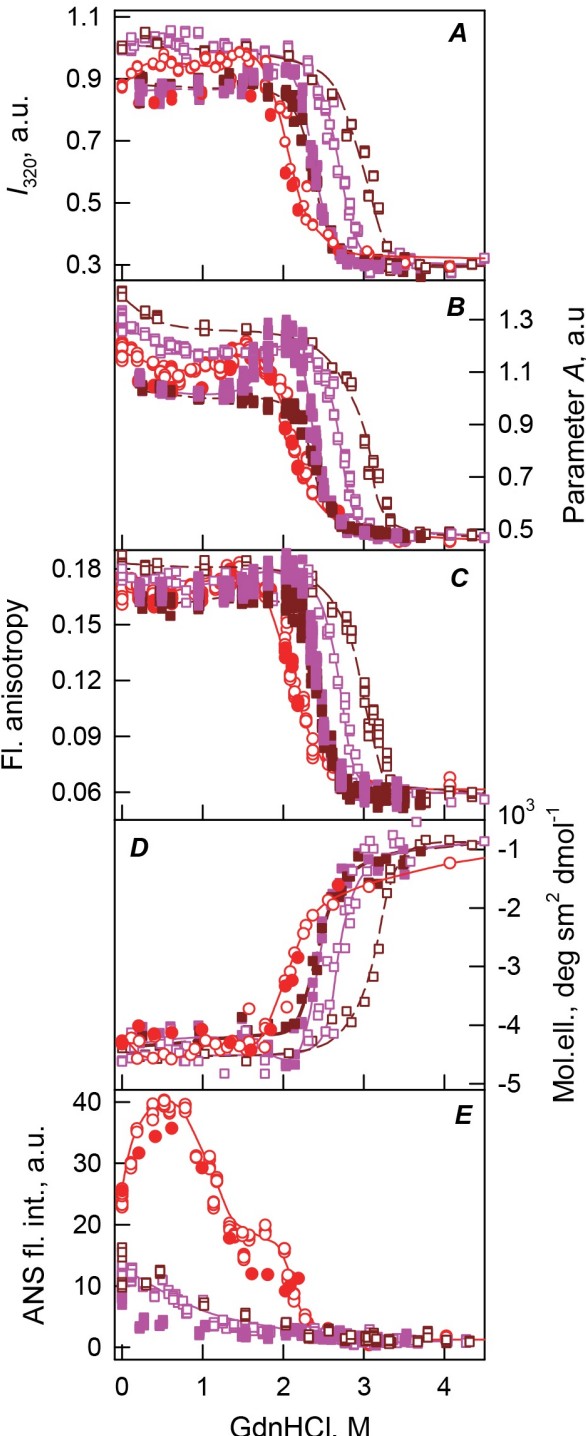

**Figure 3** **bOBP and bOBP/OCT conformational changes induced by GdnHCl.** (A): changes in fluorescence intensity at 320 nm, $\lambda_{ex} = 297$ nm; (B): changes in parameter ($A$), $\lambda_{ex} = 297$ nm; (C): changes in fluorescence anisotropy at the emission wavelength 365 nm, $l_{ex} = 297$ nm; (D): changes in the ellipticity at 222 nm; (E): changes in the ANS fluorescence intensity at $\lambda_{ex} = 365$ nm, $\lambda_{em} = 480$ nm. The statistical errors for fluorescence measurements were assessed and were shown to fall 

**Figure 3 (…continued)**
within the range of 0.2–1%. The measurements were preceded by incubating the protein in a solution with the appropriate GdnHCl concentration at 4 °C for 24 (red circles) in the case of bOBP. The open symbols indicate unfolding, whereas the closed symbols represent refolding. While studying the folding of bOBP/OCT (squares), the solution of complex of the protein with its ligand were incubated in a solution with the appropriate GdnHCl concentration at 4 °C for less than 24 h (open brown squares), up to 120 h (open pink squares) at the protein denaturation, and 1 h (closed brown squares) and 72 h–30 days (closed pink squares) at the protein renaturation.

This hydrophobic fluorescent probe is frequently used for the analysis of the presence of solvent-exposed hydrophobic patches in a protein (*Stryer, 1965*) and for the detection of accumulation of partially folded intermediates during equilibrium and kinetic protein unfolding-refolding processes due the ability of ANS to bind to such solvent-exposed hydrophobic patches (which are commonly found in partially folded proteins) and due to the fact that this interaction can be easily detected by the significant increase in the ANS fluorescence intensity and a characteristic blue-shift of its fluorescence maximum (*Semisotnov et al., 1991*). The shape of the unfolding curve monitored by the GdnHCl-induced changes in the fluorescence intensity of ANS added to the bOBP/OCT was remarkably different from the unfolding curve measured for the recombinant bOBP alone. We observe a smooth continuous decrease in the ANS fluorescence intensity at moderate GdnHCl concentrations, with the ANS fluorescence intensity reaching zero at the denaturant concentrations leading to the formation of the compact dimeric form of bOBP (Fig. 3).

These observations suggest that ANS interacts with bOBP at sites close to and/or overlapping with the ligand binding sites. Therefore, the formation of the bOBP/OCT complex prevents ANS binding. Earlier analysis of the dimeric bOBP structure revealed the presence of an additional ligand binding site at the interface between the monomeric subunits (*Bianchet et al., 1996*; *Ikematsu, Takaoka & Yasuda, 2005*; *Pevsner et al., 1985*). However, this inter-subunit binding site was shown to be noticeably weaker than the major ligand binding site located within the β-barrel (*Bianchet et al., 1996*; *Ikematsu, Takaoka & Yasuda, 2005*; *Pevsner et al., 1985*). Our data agree with the presence of an additional ligand binding site in a protein. At the formation of the dimeric bOBP/OCT complex with the native-like compactness at 2.0 M GdnHCl, this additional site is occupied by the ligand, also preventing its interaction with ANS.

Moderate ANS fluorescence is detected in solutions containing less than 2 M GdnHCl; i.e., under conditions where the bOBP/OCT complex exists as a mixture of monomeric and dimeric molecules, which are different from the native dimeric form of the bOBP. Under these conditions, ANS fluorescence intensity in the presence of the bOBP/OCT complex is noticeably lower than the ANS fluorescence recorded for the bOBP alone. These observations suggest that under these conditions the additional ligand binding site of the dimeric bOBP/OCT complex is occupied by ANS, whereas the inner cavity of the barrel is engaged in ligand binding. It is likely that the inability of the natural ligand to interact with the additional weak ligand binding site located between the monomeric

subunits can be due to the structural difference of this site in the native dimeric bOBP and in a protein in the original native-like state or an intermediate compact state.

Analysis of the bOBP/OCT refolding from the completely unfolded state revealed that the dependencies of various structural characteristics of the bOBP/OCT on GdnHCl concentrations depend on the incubation time of this complex in the presence of the denaturant (see Fig. 3). In fact, during the refolding process, equilibrium values of the analyzed structural characteristics of the bOBP/OCT complex are reached after the incubation of this complex in the presence of the desired GdnHCl concentration for 72 h. No subsequent changes were detected when protein was incubated for 30 days. This analysis revealed the presence of noticeable hysteresis between the curves describing the equilibrium unfolding and refolding of the bOBP/OCT complex in a wide range of the GdnHCl concentrations. In fact, the equilibrium unfolding and refolding curves coincide only in the vicinity of 2.0 M GdnHCl, where, according to the gel-filtration analysis, the native dimeric state of the bOBP/OCT complex is formed, whereas within the region corresponding to the transition from the native dimeric form to the completely unfolded state of the bOBP/OCT complex, equilibrium curves describing unfolding and refolding of this complex do not coincide.

The equilibrium refolding curve describing transition from the unfolded to the compact dimeric state of the bOBP/OCT complex is shifted toward the lower GdnHCl concentrations in comparison with the equilibrium unfolding curve (Fig. 3). However, in comparison with the unfolding of the bOBP alone, this equilibrium refolding curve of the bOBP/OCT complex is still shifted toward higher GdnHCl concentrations. These data suggest that at the same denaturant concentrations, the fractions of native bOBP formed during the refolding from the completely unfolded state are significantly lower than the fraction of native protein remaining within the region of the bOBP/OCT unfolding. However, once formed, the native protein gains the ability to bind ligand. This hypothesis is supported by the results of the gel-filtration analysis (Fig. 2). For example, the elution profiles registered during the unfolding and refolding of the bOBP/OCT complex at 2.5 M GdnHCl show that under these conditions, more native protein is present during the bOBP/OCT unfolding, whereas unfolded species prevail during the refolding of this complex. Therefore, the effective rates of the formation of various bOBP conformers are significantly different during the unfolding and refolding processes and noticeably depend on the denaturant concentration.

It is likely that the same reasons define the irreversibility of the unfolding of the bOBP/OCT complex at low GdnHCl concentrations. Under these conditions, the rate of the formation of the monomeric bOBP/OCT complex is significantly higher than the rate of the dimeric bOBP/OCT formation. As a result, refolding of the bOBP/OCT complex at the low GdnHCl concentrations results in the preferential formation of monomeric bOBP/OCT species, whereas under the identical conditions, the unfolding reaction mixture contains roughly equimolar quantities of the bOBP/OCT monomers and dimers (Figs. 2 and 3).

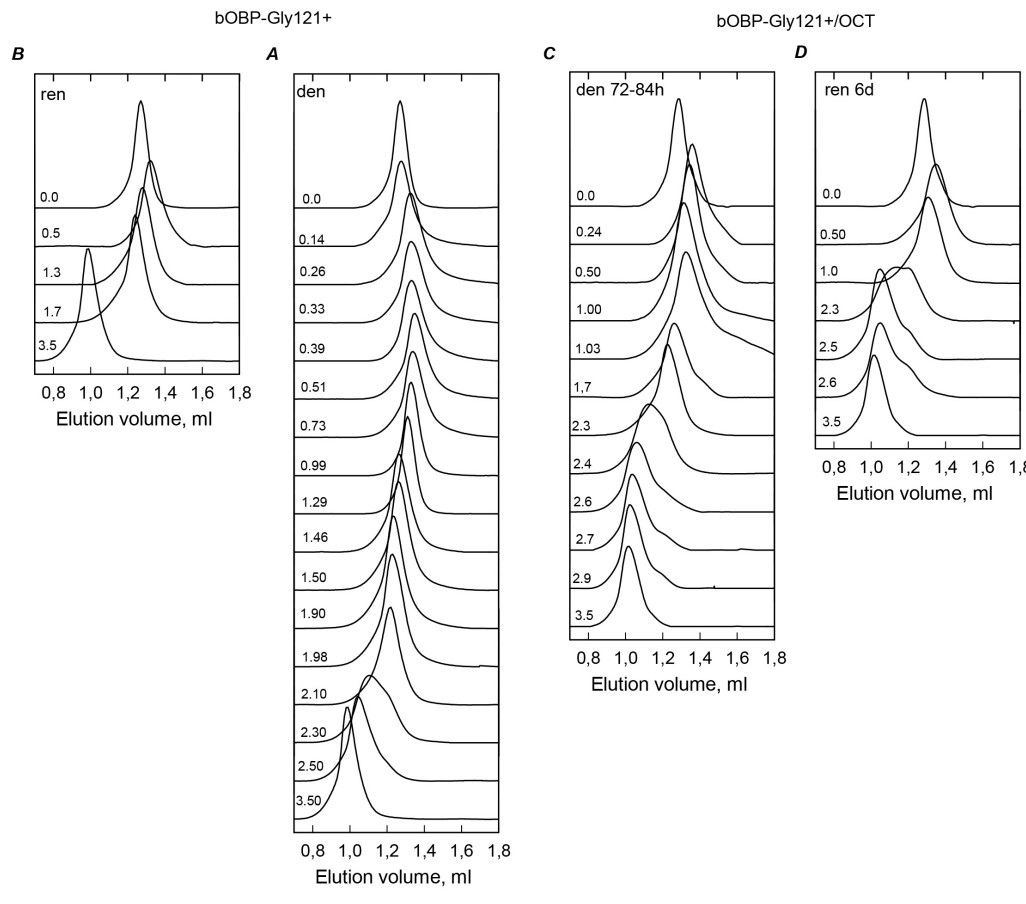

**Figure 4** **The changes of hydrodynamic dimensions of recombinant bOBP-Gly121+ (A and B) and its complex with ligand bOBP-Gly121+/OCT (C and D) in different structural states.** The elution profiles for bOBP-Gly121+ and bOBP-Gly121+/OCT were recorded during the protein denaturation (A and C) and renaturation from unfolded states (B and D) induced by GdnHCl. The elution profiles for bOBP-Gly121+ were measured after pre-incubation of the protein and the solution of GdnHCl in desired concentration for 24 h (A and B), while in the case of bOBP-Gly121+/OCT the incubation time was extended to 72–84 h for denaturation (C) and 6 days for renaturation (D). The figures on the curves are the GdnHCl concentrations.

## Equilibrium unfolding of the monomeric bOBP-Gly121+

Already at relatively low GdnHCl concentrations, the monomeric bOBP-Gly121+ is converted to the compact partially folded state with structural characteristics resembling those of the partially folded species accumulated during the equilibrium unfolding of the recombinant bOBP (see Figs. 4, 5 and Table 1). This compact intermediate is able to bind ANS and exists in a wide range of the GdnHCl concentrations (up to about 1.3 M GdnHCl). Subsequent increase in the denaturant concentration promotes transition to a more loose form, which, at the further increase of the GdnHCl concentration, is converted to the completely unfolded state. This GdnHCl-induced unfolding of the bOBP-Gly121+ is a completely reversible process as evidenced by the coincidence of the equilibrium characteristics of the protein measured at the processes of the bOBP-Gly121+ unfolding and refolding.

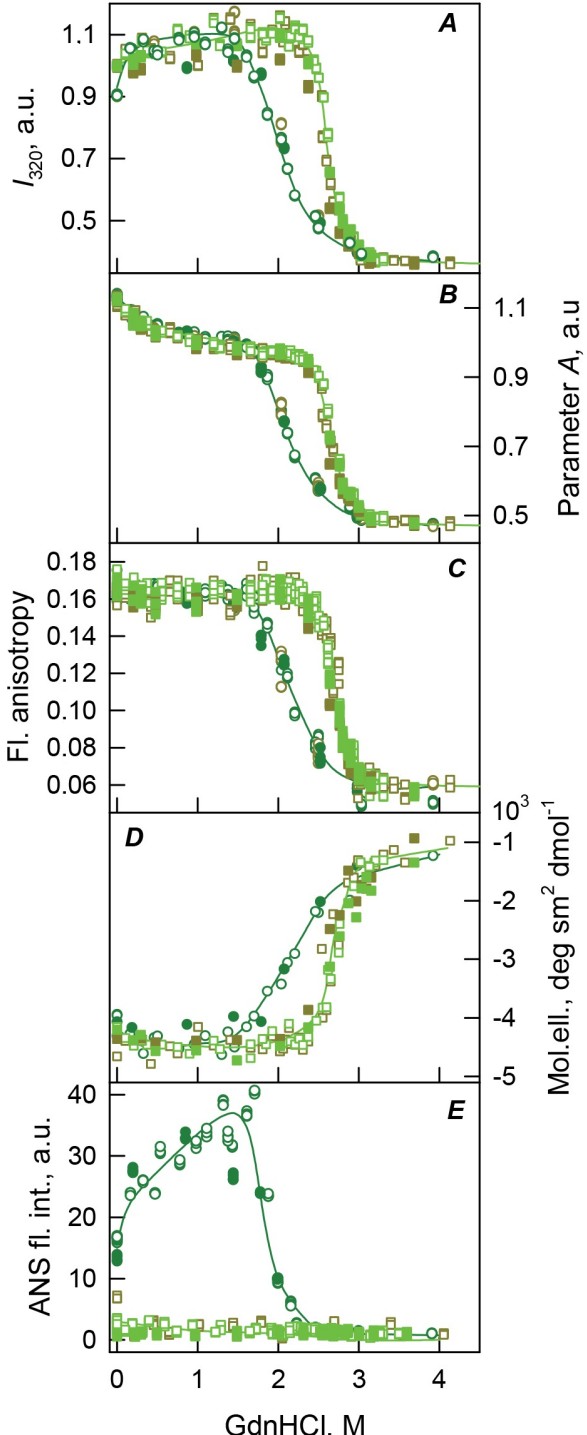

**Figure 5** **bOBP-Gly121+ and bOBP-Gly121+/OCT conformational changes induced by GdnHCl.** (A): changes in fluorescence intensity at 320 nm, $\lambda_{ex} = 297$ nm; (B): changes in parameter ($A$), $\lambda_{ex} = 297$ nm; (C): changes in fluorescence anisotropy at the emission wavelength 365 nm, $l_{ex} = 297$ nm; (D): changes in the ellipticity at 222 nm; (E): changes in the ANS fluorescence intensity at $\lambda_{ex} = 365$ nm, $\lambda_{em} = 480$ nm. The measurements were preceded by incubating the protein in a solution (continued on next page...)

 

The formation of the bOBP-Gly121+/OCT complex results in a noticeable stabilization of this protein. This is evidenced by the increase in the cooperativity of the unfolding transition, which is also shifted toward higher GdnHCl concentrations. However, the formation of a complex between the bOBP-Gly121+ and OCT does not affect the unfolding mechanism of this protein.

## GdnHCl-induced unfolding of the monomeric GCC-bOBP

Analysis of the peculiarities of the equilibrium unfolding and refolding processes monitored by the GdnHCl-induced changes in various structural characteristics of the monomeric GCC-bOBP suggests that the unfolding of this protein is a completely reversible process accompanied by the formation of partially folded intermediates similar to those observed during the equilibrium unfolding of the recombinant bOBP and its monomeric bOBP-Gly121+ form (see Figs. 6, 7 and Table 1). However, although qualitatively unfolded processes of these three proteins are similar, there are some noticeable differences. For example, in comparison with the recombinant bOBP and bOBP-Gly121+ unfolding, a compact intermediate with high ANS affinity is formed at higher denaturant concentrations during the GCC-bOBP unfolding (at 1.0 M GdnHCl). This illustrates higher conformational stability of the disulfide-stabilized GCC-bOBP compared to the recombinant bOBP and its monomeric form bOBP-Gly121+.

GCC-bOBP is further stabilized due to the GCC-bOBP/OCT complex formation. Refolding curves detected by changes in different structural characteristics of this complex and registered after the incubation of the corresponding solutions for one hour coincide with the transition curves describing the equilibrium unfolding of GCC-bOBP, and subsequent incubation of these same solutions for 72 h leads to the detectable shift of the transition curves. As a result, equilibrium unfolding and refolding transitions of the GCC-bOBP/OCT complex coincide suggesting that the unfolding of this protein is a completely reversible process. However, GCC-bOBP becomes able to bind ligand only after the formation of correct native structure stabilized by the disulfide bond. Earlier similar effects were described for other ligand-binding protein, such as the D-glucose/D-galactose-binding protein (GGBP) from *E. coli* (*Stepanenko et al., 2011a*; *Stepanenko et al., 2009*; *Stepanenko et al., 2011b*). In fact, our analysis of the peculiarities of the GGBP unfolding revealed that ligand binding might constitute a rate-limiting stage of the protein unfolding-refolding process. This phenomenon can be understood considering the fact that the formation of the protein-ligand complex depends on the appearance of the matching configurations between the ligand and the active site of a fully formed native protein (*Stepanenko et al., 2011a*; *Stepanenko et al., 2009*; *Stepanenko et al., 2011b*).

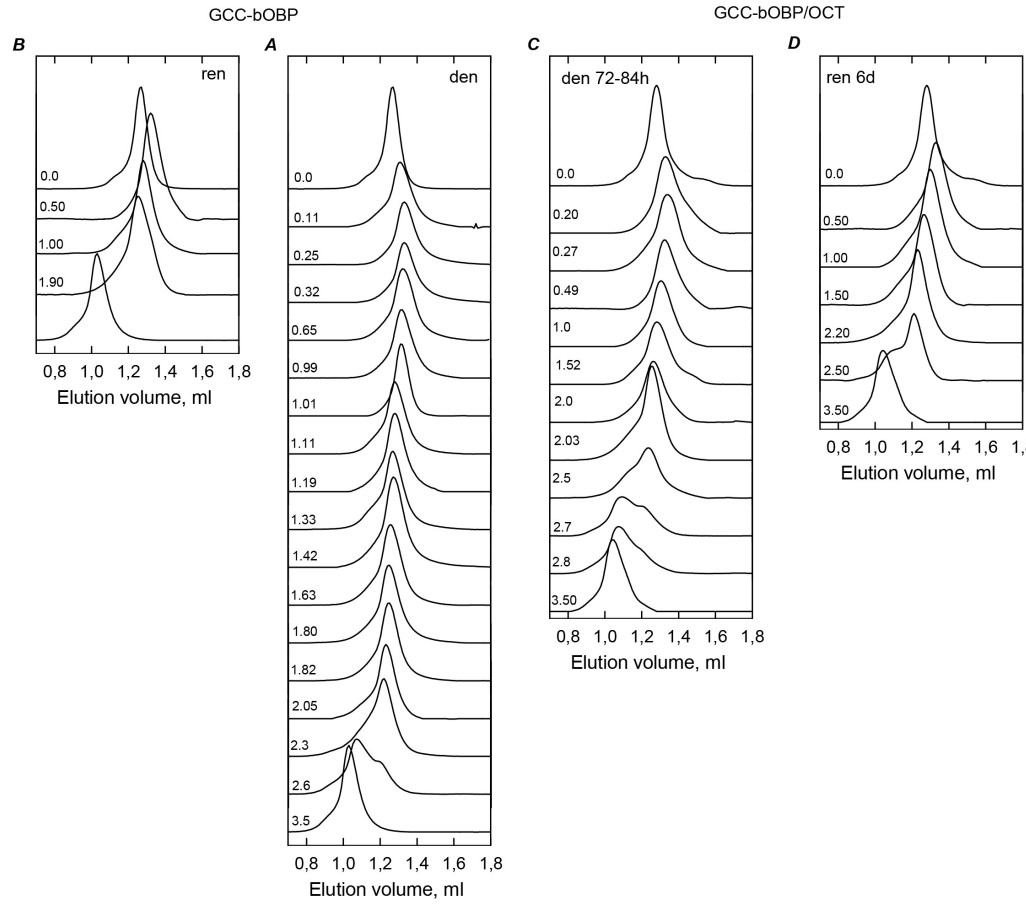

**Figure 6** **The changes of hydrodynamic dimensions of recombinant GCC-bOBP (A and B) and its complex with ligand GCC-bOBP/OCT (C and D) in different structural states.** The elution profiles for GCC-bOBP and GCC-bOBP/OCT were recorded during the protein denaturation (A and C) and renaturation from unfolded states (B and D) induced by GdnHCl. The elution profiles for GCC-bOBP were measured after pre-incubation of the protein and the solution of GdnHCl in desired concentration for 24 h (A and B), while in the case of GCC-bOBP/OCT the incubation time was extended to 72–84 h for denaturation (C) and 6 days for renaturation (D). The figures on the curves are the GdnHCl concentrations.

Our current analysis revealed that the unfolding of the monomeric complexes bOBP-Gly121+/OCT and GCC-bOBP/OCT is not accompanied by the ANS fluorescence enhancement in the whole range of GdnHCl concentrations (see Figs. 5 and 7). These observations support the hypothesis on the existence of the additional ligand-binding site in the dimeric bOBP.

Table 2 shows that the equilibrium unfolding transition recorded for the recombinant bOBP coincides with that of its monomeric bOBP-Gly121+ form. On the other hand, unfolding transition of the monomeric GCC-bOBP stabilized by the engineered disulfide bond is noticeably shifted to higher denaturant concentrations. Curiously, the unfolding of complexes of all proteins analyzed in this study with natural ligand OCT happens at the same denaturant concentrations.

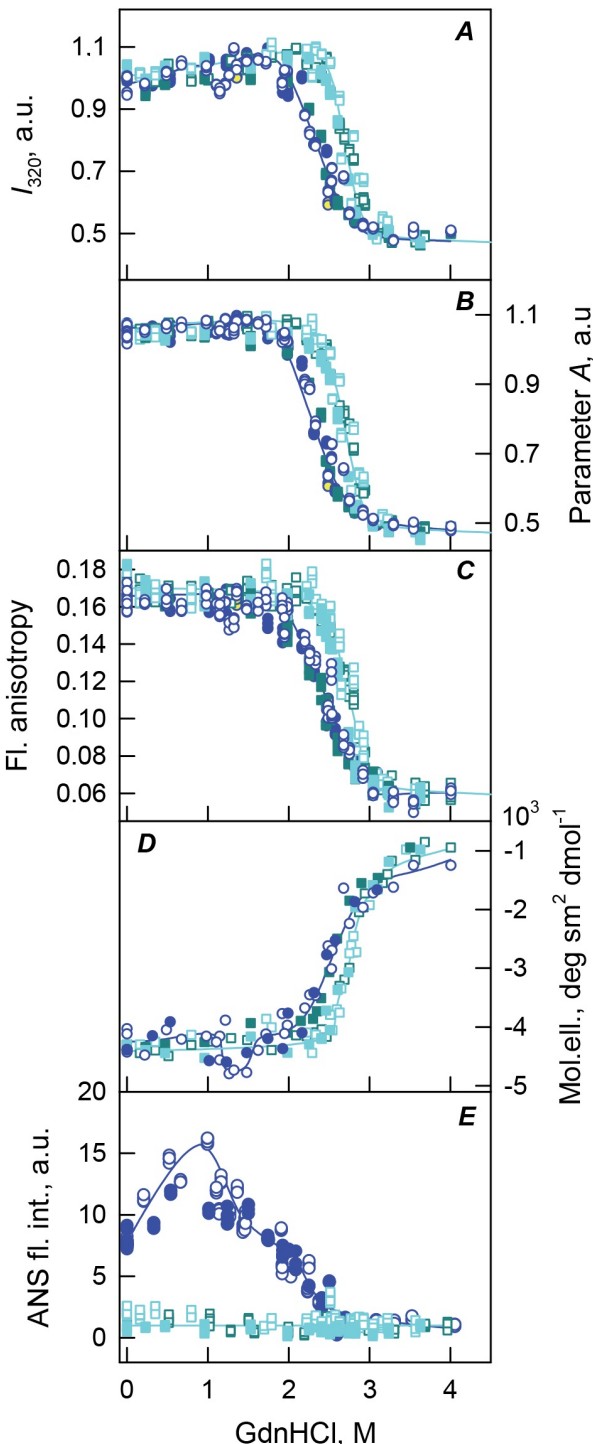

**Figure 7** **GCC-bOBP and GCC-bOBP/OCT conformational changes induced by GdnHCl.** (A) Changes in fluorescence intensity at 320 nm, $\lambda_{ex} = 297$ nm; (B) changes in parameter (A), $\lambda_{ex} = 297$ nm; (C) changes in fluorescence anisotropy at the emission wavelength 365 nm, $l_{ex} = 297$ nm; (D) changes in the ellipticity at 222 nm; (E): changes in the ANS fluorescence intensity at (continued on next page…)

**Figure 7 (…continued)**
$\lambda_{ex} = 365$ nm, $\lambda_{em} = 480$ nm. The measurements were preceded by incubating the protein in a solution with the appropriate GdnHCl concentration at 4 °C for 24 (blue circles) in the case of GCC-bOBP. The open symbols indicate unfolding, whereas the closed symbols represent refolding. While studying the folding of GCC-bOBP/OCT (squares), the solution of complex of the protein with its ligand were incubated in a solution with the appropriate GdnHCl concentration at 4 °C for less than 24 h (open dark blue squares), up to 72 h (open light blue squares) at the protein denaturation, and 1 h (closed dark blue squares) and 72 h (closed light blue squares) at the protein renaturation.

**Table 2** Thermodynamic parameters of GdnHCl-induced denaturation of bOBP, its mutant variants and their complexes with octen-3-ol (OCT).

| Protein | $m$ (kJ mol$^{-1}$ M$^{-1}$) | $C_m$ (M)[a] | $\Delta G^0_{N-U}$ [b] (kJ mol$^{-1}$) |
|---|---|---|---|
| bOBP | 3.7±0.2 | 2.1±0.1 | 7.7±0.6 |
| bOBP/OCT | 4.1±0.2 | 2.7±0.1 | 11.0±0.5 |
| bOBP/Gly121+ | 2.8±0.2 | 2.0±0.1 | 5.5±0.4 |
| bOBP/Gly121+/OCT | 5.3±0.2 | 2.6±0.1 | 14.0±0.5 |
| GCC-bOBP | 2.5±0.3 | 2.3±0.1 | 5.8±0.7 |
| GCC-bOBP/OCT | 4.5±0.3 | 2.7±0.1 | 12.0±0.9 |

**Notes.**
[a] $C_m$ is the denaturant concentration at midpoint of conformational transition.
[b] The fluorescence signals of the folded and unfolded states were approximated by linear dependences as function of denaturant concentration (*Nolting, 1999*).

It is important to note here that the quantitative characterization of the affinity of the bOBP and its mutants to the natural OCT ligand is beyond the scope of this study. In fact, the binding constant of the protein was determined in previous studies (*Bianchet et al., 1996*; *Ikematsu, Takaoka & Yasuda, 2005*; *Pevsner et al., 1985*). According to published data, binding constant of a natural OCT ligand to the triple mutant GCC-bOBP is in the micromolar range, which is similar to the affinity of the wild type protein (*Ramoni et al., 2008*). Amino acid substitutions introduced in mutant proteins, namely the Gly 121+ insertion and the W64C and H155C replacements, do not affect the ligand-binding site of the protein. Data reported in the first paper of this series (*Stepanenko et al., 2016*) revealed that these substitutions do not have significant influence on the protein tertiary and secondary structure. All these data indicate that the mutant forms of the bOBP protein might retain affinity inherent to the wild-type protein.

Our data suggest that protein dimerization via the domain-swapping mechanism does not contribute much to the increase in the conformational stability of a protein (at least in the case of the analyzed in this study bOBP), despite the fact that the increased conformational stability was proposed as one of the factors determining the use of this mechanism for dimer and higher oligomer formation (*Bennett, Schlunegger & Eisenberg, 1995*; *Liu & Eisenberg, 2002*). In contrast, introduction of a disulfide bond to the structure of a monomeric protein shows significant stabilizing effects. Our data also show that the formation of a protein-ligand complex leads to the significant stabilization of different variants of bOBP and eliminates the original difference in conformational stability caused by their structural differences.

### Funding
This work was supported in part by the Program "Molecular and Cell Biology" of the Russian (K.K.T.). The funders had no role in study design, data collection and analysis, decision to publish, or preparation of the manuscript.

### Competing Interests
Irina M Kuznetsova, Vladimir N Uversky and Konstantin K Turoverov are Academic Editors for PeerJ.

### Author Contributions
- Olga V. Stepanenko and Vladimir N. Uversky conceived and designed the experiments, performed the experiments, analyzed the data, wrote the paper, prepared figures and/or tables, reviewed drafts of the paper.
- Denis O. Roginskii and Olesya V. Stepanenko performed the experiments, analyzed the data, prepared figures and/or tables, reviewed drafts of the paper.
- Irina M. Kuznetsova conceived and designed the experiments, analyzed the data, prepared figures and/or tables, reviewed drafts of the paper.
- Konstantin K. Turoverov conceived and designed the experiments, analyzed the data, reviewed drafts of the paper.

### Data Availability
All the data obtained during this study are reported in the manuscript in the form of figures and tables.

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
