# Peer review of "Structure and stability of recombinant bovine odorant-binding protein: II. Unfolding of the monomeric forms"

_PeerJ, doi:10.7717/peerj.1574_

## Round 0.1 · original submission · Major Revisions

Please address all points mentioned by the reviewers and ensure that text overlap between this manuscript and the other submissions in this series is removed.

Reviewer 1 ·

Basic reporting

The submitted manuscript by Stephanenko OV et al “Structure and stability of recombinant bovine odorant-binding protein: II. Unfolding of the monomeric forms” conforms to PeerJ policies, is well written with sufficient introduction and contains sufficiently referenced background literature. In fact large parts of the introduction and methods hardly differ from the recent publications of the same authors (Stepanenko OV, Stepanenko OV, Staiano M, Kuznetsova IM, Turoverov KK, and D'Auria S. 2014c. The quaternary structure of the recombinant bovine odorant-binding protein is modulated by chemical denaturants. PLoS One 9:e85169. And Stephanenko OV, Roginskii DO, Stephanenko OV, Kuznetsova IM, Uversky VN, and Turoverov KK. 2015. Structure and stability of recombinant bovine odorant-binding protein: I. Design and analysis of monomeric mutants. . PeerJ.) the latter not indicated as submitted only in the same journal and not published yet.
The structure of the article conforms to PeerJ policies and the information included is sufficient to make the arguments stated. It is indicated though that quite a lot of the experimental information has already been presented in the above referenced recent publications and either have to be omitted or clearly indicated that have been published elsewhere. In fact the recently submitted paper to PeerJ by the same authors has the same experimental section and a common great percentage of results.
The recommendation is that this work contains useful scientific information and should be published after considerable modification to avoid repetition of results given by the same authors in recent publications and minor presentation changes.

Experimental design

The structure of the experimental design is valid and sufficiently described and is supportive of the conclusions drawn. However most of this information has been given in the same detail in recent previous publications by the same authors and should be just referenced and not included in detail. Expressions like “the necessary GdnHCl concentration” (line127) should be replaced with the absolute range of concentration (given in Table2?). The same applies to the figures and Tables where information displayed has been duplicated.

Validity of the findings

The arguments analyzed in the Discussion section is closely related to the experimental findings although curve fitting in Figure 3, 5,6 and 7 is done with no indication of statistical error. In fact Figure6 is also given as Figure4 and should be corrected. The argument that FlexPred or PONDRVSL2B could predict or distinguish changes in flexibility or intrinsic disorder with accuracy for single residue mutations and therefore since the mutants show the same profile, is an indication of no change in intrinsic disorder of flexibility, is not firmly based with no atomic experimental evidence and should be further supported.

Additional comments

Minor corrections

Line 77 “effect of the environmental feature” should be rephrased
Line 254 “occupies” change to “occupied”
Line 254 “the ligand binding” omit the “the”
Line 294 “dimmer” replace with “dimer”
Line 490 “PeerJ.” replace with “PeerJ. Submitted”
Line 575 “Figure4” omit

·

Basic reporting

The paper presents interesting information but several aspects can be improved. Please see my comments for authors.

Experimental design

Binding of OCT to the OBP should be measured independently and be quantitative

Validity of the findings

The findings are valid but limited, additional quantitative information should be given

Additional comments

In this study the authors analyze the conformational stability of recombinant bovine odorant-binding protein (bOBP) and a set of mutants including the monomeric bOBP-Gly121+ and a mutant with an additional intramolecular disulfide bond present in OBP’s from other species. They also investigated the effect of the natural ligand OCT on the stability of the protein. They have used steady-state fluorescence (the fluorescence lifetimes present in table 1 do not really add relevant information to the paper), CD and gel filtration to characterize protein stability and dimer/monomer transition.

Some points in the work need additional insight/discussion.
1. The unfolding of the secondary structure occurs at lower GdnHCl concentrations (fig 2D) compared to unfolding of the tertiary structure (figure 2A and B). This is not a typical behavior and I would like to see some discussion regarding this point.
2. The binding of OCT to bOBP was just inferred from stability profiles, gel filtration experiments and ANS fluorescence and not measured directly. May I suggest that the authors use an independent technique such as ITC to characterize this binding and determine affinity constants?
2. Indeed, this work needs further quantitative data besides the qualitative evaluation provided by unfolding/refolding profile plots. Figure 9 is just a repetition of the data provided in previous figures so I think that the authors can compare the stability of the different mutants of bOBP in a table (instead of figure 9) where they can present quantitative stability parameters such as DeltaG in water, cooperativeness and mid-point of transition(s). The stability of non two-state unfolding/refolding proteins such as bOBP can also be quantitative.

---

## Round 0.2 · Minor Revisions

I am satisfied with the revisions, which I believe address the reviewers' comments. I must ask you, however, to double-check the derivation of the formulas for the fitting of the denaturation curves: your formula (1) (line 139) seems to imply K=Fu/Fn, instead of K=Fn/Fu (as you state in line 141), which would mean that the signs of the DeltaG should be inverted. Please also clarify the need for constants bn and bu, and add their values (as well as those of au and an) to table 2.

Reviewer 1 ·

Basic reporting

corrected

Experimental design

corrected

Validity of the findings

corrected

Additional comments

Most suggestions have been incorporated in the text and explanations given to a satisfactory level.

---

## Round 0.3 · accepted · Accept

The manuscript has been carefully revised to satisfaction.